# Tri-Functional Calcium-Deficient Calcium Titanate Coating on Titanium Metal by Chemical and Heat Treatment

**Seiji Yamaguchi \*, Phuc Thi Minh Le, Morihiro Ito** **, Seine A. Shintani and Hiroaki Takadama**

Department of Biomedical Sciences, College of Life and Health Sciences, Chubu University, Aichi 487-8501, Japan
\* Correspondence: sy-esi@isc.chubu.ac.jp; Tel.: +81-568-51-6420

**Abstract:** The main problem of orthopedic and dental titanium (Ti) implants has been poor bone-bonding to the metal. Various coatings to improve the bone-bonding, including the hydroxyapatite and titania, have been developed, and some of them have been to successfully applied clinical use. On the other hand, there are still challenges to provide antibacterial activity and promotion of bone growth on Ti. It was shown that a calcium-deficient calcium titanate coating on Ti and its alloys exhibits high bone-bonding owing to its apatite formation. In this study, Sr and Ag ions, known for their promotion of bone growth and antibacterial activity, were introduced into the calcium-deficient calcium titanate by a three-step aqueous solution treatment combined with heat. The treated metal formed apatite within 3 days in a simulated body fluid and exhibited antibacterial activity to *Escherichia coli* without showing any cytotoxicity in MC3T3-E1 preosteoblast cells. Furthermore, the metal slowly released 1.29 ppm of Sr ions. The Ti with calcium-deficient calcium titanate doped with Sr and Ag will be useful for orthopedic and dental implants, since it should bond to bone because of its apatite formation, promote bone growth due to Sr ion release, and prevent infection owing to its antibacterial activity.

**Keywords:** antibacterial activity; bone growth; apatite formation; titanium; silver; strontium; calcium titanate; ion release; cytotoxicity; controlled release

---

## 1. Introduction

Titanium metal (Ti) and its alloy are widely used for orthopedic and dental implants since they fulfill certain clinical needs from the point of view of mechanical properties, durability, and biocompatibility. Osseointegration occurs when the surfaces of the metals were roughened at micrometer scale such as 0.5–2.0, or 3.6–5.6 μm in calculated average roughness $R_a$, and 43–50 μm in maximum height $R_z$ by plasma spraying, grid blasting and/or acid etching [1,2]. The nanometer-scale roughness produced by anodic oxidation also has been found to increase cell adhesion, proliferation and alkaline phosphatase activity [3,4]. Although these roughened Ti surfaces are able to directly contact with living bone, they still do not bond to it adequately. Mineralization process is a promising method to achieve strong and stable bone-bonding. It has been reported that bioactive glass/ceramics such as bioglass, hydroxyapatite, and glass ceramics A-W directly bonded to bone through the bone-like apatite layer formed on their surfaces [5].

Various types of surface coating of bioactive glass/ceramics by plasma spray, sputtering, sol-gel, and alternative soaking have been attempted [6–8]. Among them, a plasma spray coating of hydroxyapatite has been widely used to confer bone-bonding to total hip joint, dental implant, and so on. However, this does not form a stable bioactive surface layer, since the surfaces of the hydroxyapatite particles exposed to the plasma are partially melted, so the resultant calcium phosphate coating is liable to be decomposed in the living body in process of time [6].

Various types of surface modifications including alkali/acid solution and heat treatment, hydrothermal treatment, and ion implantation have been developed to confer apatite-forming capability on the metals so that the activated metals form bone-like apatite spontaneously on their surfaces by using the calcium and phosphate present in body fluids, and thereby bond to bone through the apatite [9–12]. Among the variety of the modification techniques, the alkali/acid solution and heat treatment has the needed characteristics for producing a uniform activated surface layer, even on the inner wall of a porous body, without requiring any especial apparatus [13,14]. It has been demonstrated that a bioactive sodium titanate layer is produced on Ti when the metal is soaked in 5 M NaOH solution at 60 °C for 24 h and subsequently heat-treated at 600 °C for 1 h [15,16]. The surface of treated metal forms a bone-like apatite spontaneously in the living body and bonds bone through this layer [16]. Total artificial hip joints (THAs) with the bioactive sodium titanate layer on their porous Ti layer have been under clinical use since 2007. A recent ten-year follow-up revealed the beneficial effects of the NaOH-heat-treated THAs to be a high survival rate (98%), no radiographic signs of loosening, and both early and stable bone-bonding [17]. However, two joints were retrieved owing to deep infection and periprosthetic femoral fracture, since the NaOH-heat-treated THAs neither promoted bone growth nor prevented infection [17]. Subsequently, the NaOH-heat treatment was modified to NaOH-CaCl$_2$-heat-water treatment to produce a calcium-deficient calcium titanate layer on Ti and Ti alloys, which resulted in more stable apatite formation and bone-bonding [18–20]. On the other hand, there are still the challenges of providing antibacterial activity and promoting bone growth on Ti. It has been reported that typically 1%–2% of patients with total hip arthroplasties suffer deep infections [21]. Dental peri-implant disease and infection have become a main focus of oral implantology [22].

Strontium (Sr) and silver (Ag) ion are candidates for the promotion of new bone formation and prevention of infection since the former exerts a therapeutic effect on osteoporosis bone due to increased new bone formation and deceased bone resorption, while the later prevents infection because of its strong effect against a broad range of microorganisms [23–26]. Pre-clinical study reports have shown that the Sr ions released from dosed strontium ranelate improve mineral density at various skeletal sites such as the total hip and lumbar spine, resulting in the improvement of osteoporosis [23,24]. The mechanism of the antimicrobial action of Ag ions is understood as resulting from an interaction with the thiol (sulfhydryl) groups in enzymes and proteins [25], and is effective even in the living body [26]. Studies have reported the separate incorporation of Sr or Ag into the surface of Ti and Ti alloys [27–30], but there are few reports of these ions being incorporated simultaneously. There are even reported attempts to incorporate these ions into the Ti surface in an effort to confer a capacity for apatite formation. It was reported that Ag-doped calcium phosphate coatings was produced on Ti by a combination of anodic oxidation, electrophoretic deposition, and magnetron-sputtering [31,32]. Although the coated metal exhibited strong antibacterial activity against *Escherichia coli* (*E. coli*), it a little decreased cell viability [32]. A novel method is desired to confer Ti the capabilities of excellent antibacterial activity without any decrease in cell viability, direct bone-bonding, and promotion of new bone formation at the same time.

In this study, Sr and Ag ions were introduced into the calcium-deficient calcium titanate produced on Ti under controlled conditions so that the treated Ti slowly released Sr and Ag ions in order to exhibit the functions of promoting new bone formation while preventing infection without decreasing apatite formation. The potential of the treated metal for clinical applications is discussed in terms of Sr and Ag ion release, antibacterial activity, cytocompatibility, and apatite formation.

## 2. Materials and Methods

### 2.1. Surface Treatment

Commercially pure Ti sections (Ti > 99.5%; Nilaco Co., Tokyo, Japan) 10 mm × 10 mm × 1 mm in size was grinded with #400 diamond plates and then cleaned in an ultrasonic bath by using acetone, 2-propanol and ultrapure water for 30 min, and dried at 40 °C overnight. They were immersed in

5 M NaOH (Reagent grade; Kanto Chemical Co., Inc., Tokyo, Japan) solution at 60 °C with shaking at 120 strokes/min for a period of 24 h followed by gentle rinse under ultrapure water flow for 30 s. The treated metals were soaked in a mixed solution consist of 50 mM $CaCl_2$ (Reagent grade; Kanto Chemical Co., Inc., Tokyo, Japan) and 50 mM $SrCl_2$ (Reagent grade; Kanto Chemical Co., Inc., Tokyo, Japan) at 40 °C, shaken at 120 strokes/min for 24 h, then washed and dried in a similar manner (designated as "Ca + Sr"). They were subsequently heated at 600 °C with programming rate of 5 °C/min and holding time of 1 h, then naturally cooled in an electric furnace. After the heat treatment, they were immersed in a mixed solution of 1 M $Sr(NO_3)_2$ (Reagent grade; Kanto Chemical Co., Inc., Tokyo, Japan) and *X* mM $AgNO_3$ (Reagent grade; Kanto Chemical Co., Inc., Tokyo, Japan) with an adjusted pH from 3 to 8 by adding $HNO_3$ or $NH_3$(aq) at 80 °C, where *X* is a range from 1 to 100 mM and designated as "Sr + *X* mM Ag, shaken , washed, and dried in the in a similar manner. The nominal and measured pH of the 1 M $Sr(NO_3)_2$ + 1 mM $AgNO_3$ are summarized in Table 1. Some of the Ti samples subjected to the same NaOH-Ca + Sr-heat treatment were subsequently soaked in 1 M $SrCl_2$ solution without Ag for comparison.

**Table 1.** Measured pH of 1 M $SrNO_3$ + 1 mM $AgNO_3$ solution corresponding to nominal pH.

| Nominal pH | Measured pH | Additive |
|:---:|:---:|:---:|
| pH = 3 | 3.06 | $HNO_3$ |
| pH = 4 | 3.90 | $HNO_3$ |
| pH = 5 | 4.80 | No additives |
| pH = 6 | 6.01 | $NH_3$(aq) |
| pH = 7 | 7.16 | $NH_3$(aq) |
| pH = 8 | 7.83 | $NH_3$(aq) |

*2.2. Surface Analysis*

2.2.1. Scanning Electron Microscopy and Energy Dispersive X-ray Analysis

The samples treated with the aqueous solution and heat were examined by field emission scanning electron microscopy (FE-SEM: S-4300, Hitachi Co., Tokyo, Japan) equipped with an energy dispersive X-ray spectrometer (EDX: EMAX-7000, Horiba Ltd., Kyoto, Japan). In FE-SEM observation, the samples were subjected to a thin-film coating of Pt–Pd, and 15 kV accelerate voltage was selected. In EDX anlaysis, the elements of Ca, Ag, O, and Ti were quantified using 9 kV. The measurement was repeated on five different points, and their averaged value was used.

2.2.2. Thin-Film X-ray Diffraction and Fourier Transform Confocal Laser Raman Spectrometry

The surface structure of the samples subjected to the aqueous solution and heat treatment were analyzed by a thin-film X-ray diffractometer (TF-XRD: model RNT-2500, Rigaku Co., Tokyo, Japan) and Fourier transform confocal laser Raman spectrometer (FT-Raman: LabRAM HR800, Horiba Jobin Yvon, Longjumeau, France). In TF-XRD, the measurement was conducted at a power of 50 kV and 200 mA. A CuK was used as X-ray source and the incident beam angle was set to 1° against the sample surface. In FT-Raman, measurement was conducted with 514.5 nm Ar laser at 16 mW of power excitation.

2.2.3. Scratch Resistance

The scratch resistance of the surface layer to the metal substrate was examined by a thin-film scratch tester (CSR-2000, Rhesca Co., Ltd., Tokyo, Japan) according to JIS R-3255. A stylus with diameter of 5 μm and spring constant of 200 g/mm was pressed into the treated metal surface under the conditions of scratch speed of 10 μm/s, loading rates of 100 mN/min, and amplitude of 100 μm. Five measurements were performed on each sample, and their averaged values were used for analysis.

### 2.2.4. X-ray Photoelectron Spectroscopy

The allocation of elements such as Ag, Sr, Ca, Ti, O and C on the treated samples was analyzed using X-ray photoelectron spectroscopy (XPS, PHI 5000 Versaprobe II, ULVAC-PHI, Inc., Kanagawa, Japan) with Ar sputtering (spattering rate was 4 nm/min as $SiO_2$ conversion). In the analysis, the X-ray source of an Al-K radiation line was used with the take-off angle at 45°.

The obtained spectra were calibrated by 284.8 eV in binding energy of C 1*s* peak that is of the surfactant $CH_2$ groups on the substrate.

### 2.3. Ion Release

The treated samples were immersed in 2 mL of fetal bovine serum (FBS) (Gibco, Thermo Fisher Scientific, Waltham, MA, USA) with gently shaken at a speed of 50 strokes/min at 36.5 °C. After predetermined periods up to 14 days, the $Sr^{2+}$ and $Ag^+$ ion concentrations in the FBS were determined by inductively coupled plasma emission spectroscopy (ICP, SPS3100, Seiko Instruments Inc., Chiba, Japan). The measurement was repeated 3 times for independently prepared samples, and their averaged values were calculated.

### 2.4. Soaking in Simulated Body Fluid (SBF)

After the aqueous solution and heat treatment the samples were immersed in 24 mL of acellular simulated body fluid (SBF) [33] that had been prepared according to ISO 23317. NaCl, $NaHCO_3$, KCl, $K_2HPO_4 \cdot 3H_2O$, $MgCl_2 \cdot 6H_2O$, $CaCl_2$ and $Na_2SO_4$ were purchased from Nacalai Tesque Inc., Kyoto, Japan and all of them were reagent grade. They were dissolved in fresh ultrapure water in this order, and their pH was adjusted exactly to 7.40 using tris(hydroxymethyl)aminomethane $(CH_2OH)_3CNH_2$ and 1 M HCl at 36.5 °C. After immersion periods of 3 days, the samples were rinsed and dried. Apatite formation formed on the metal surface was examined using FE-SEM, TF-XRD, and EDX.

### 2.5. Antibacterial Activity Test

The antibacterial activity of the treated Ti samples was evaluated by the film contact method (ISO22196) [34]. An *E. coli* (IFO 3972) suspension of 100 μL was inoculated on the treated Ti samples at 25 mm × 25 mm × 1 mm and then covered with a 20 mm × 20 mm polypropylene film that had been sterilized with ethanol and dried for 7 days in a clean bench. They were placed in a 100 mm diameter petri dish with a sterilized plastic cap filled with sterilized pure water to prevent drying of the bacterial suspension, and then stored in an incubator under 95% relative humidity at 35 °C for 24 h. After incubation, each sample was washed with 10 mL of a soybean casein digest broth containing lecithin and polyoxyethylene sorbitan monooleate (SCDLP broth) to collect the bacteria. The recovered suspension was subjected to ten-fold serial dilutions, followed by placed in petri dishes containing standard plate count agar at 35 °C for 48 h. After incubation, the number of viable *E. coli* was calculated using the dilution factor and the number of colonies that was counted on the petri dish. Finally, the antibacterial activity value (*R*) was calculated for each specimen as follows:

$$R = \{\log(B/A) - \log(C/A)\} = \log(B/C) \tag{1}$$

where *A* and *B* are the numbers of viable *E. coli* recovered from the untreated specimen immediately or 24 h incubation after inoculation. *C* is the number of viable *E. coli* recovered from the treated specimen immediately after 24 h incubation.

### 2.6. Cell Proliferation

MC3T3-E1 cells (subclone 14, ATCC, Manassas, VA, USA) were seeded onto Ti disk specimens that were 18 mm in diameter in 12-well plates at a density of $2 \times 10^4$ cells/well. They were cultured in $\alpha$-MEM (Gibco, Thermo Fisher Scientific, Waltham, MA, USA) with 10% FBS and 1% penicillin/streptomycin at

37 °C in 5% $CO_2$ atmosphere. After 1 and 3 days, the cell count reagent SF (Nacalai tesque, Kyoto, Japan) was added to the medium and stored in the incubator for 2 h. After the incubation, 100 μL of the medium was taken to a 96-well plate. The absorbance at 450 nm that is attributed to formazan product derived from living cells was quantified by a Microplate reader (iMark$^{TM}$, Bio-Rad, Hercules, CA, USA). Four disk specimens were prepared for each sample type in this measurement.

## 2.7. Statistical Analysis

The obtained data in Section 2.6 was statistically analyzed by out using R language with these libraries (mvtnorm, survival, MASS, TH. data, multcomp, abind). The sample group data were initially tested for normality (Kolomogorov-Smirnov test) and homoscedasticity of variance (Bartlett's test). One-way analysis of variance (ANOVA) was adopted in the groups that satisfy those conditions to find any significant differences in the measured variables between control and treatment groups. When a difference was detected (*p*-value < 0.05), Tukey's multiple comparison test was performed to identify which treatment groups were significantly different. In this case, the ANOVA was satisfied in all analyses.

## 3. Results

### 3.1. Effect of the pH of the Solution Used on Apatite Formation

The chemical composition of the Ti surface after each aqueous solution and heat treatment was analyzed by EDX analysis. As shown in Table 2, 5.1% Na was induced by the initial NaOH treatment, and then replaced with 2.2% Ca and 1.3% Sr by the subsequent Ca + Sr treatment, which remained after the heat treatment. When the treated metal was immersed in 1 M $SrCl_2$ solution, amount of Sr a little increased, probably due to the additional induce of Sr into the surface because of high concentration of Sr ions in the solution. When the treated Ti specimens were soaked in 1 M $Sr(NO_3)_2$ and 1 mM $AgNO_3$ with a pH equal to or less than 4, 0.2% of the Ag was introduced into the surface, while the Ca amount was slightly decreased. In contrast, no decrease in the Ca content was observed when the Ti specimens were soaked in a solution with a pH greater than 4. The amount of Ag intoroduced into the Ti surface tended to decrease with an increase in the pH of the solution.

**Table 2.** The results of EDX analysis on the surface of Ti subjected to Sr + 1 mM Ag treatment with various pH following NaOH, Ca + Sr and heat treatment.

| Treatment | Element/at.% | | | | | |
|---|---|---|---|---|---|---|
| | O | Ti | Na | Ca | Sr | Ag |
| NaOH | 65.1 | 29.8 | 5.1 | 0 | 0 | 0 |
| NaOH-Ca + Sr | 68.1 | 28.4 | 0 | 2.2 | 1.3 | 0 |
| NaOH-Ca + Sr-heat | 68.9 | 27.6 | 0 | 2.3 | 1.3 | 0 |
| NaOH-Ca + Sr-heat-Sr + 1 mM Ag (pH = 3) | 66.8 | 29.8 | 0 | 1.8 | 1.4 | 0.2 |
| NaOH-Ca + Sr-heat-Sr + 1 mM Ag (pH = 4) | 65.8 | 30.5 | 0 | 1.9 | 1.6 | 0.2 |
| NaOH-Ca + Sr-heat-Sr + 1 mM Ag (pH = 5) | 66.1 | 30.0 | 0 | 2.2 | 1.6 | 0.2 |
| NaOH-Ca + Sr-heat-Sr + 1 mM Ag (pH = 6) | 66.3 | 29.8 | 0 | 2.1 | 1.6 | 0.2 |
| NaOH-Ca + Sr-heat-Sr + 1 mM Ag (pH = 7) | 65.6 | 30.6 | 0 | 2.1 | 1.6 | 0.1 |
| NaOH-Ca + Sr-heat-Sr+1 mM Ag (pH = 8) | 65.7 | 30.4 | 0 | 2.1 | 1.7 | 0.1 |
| NaOH-Ca + Sr-heat-Sr | 68.7 | 27.4 | 0 | 2.2 | 1.7 | 0 |

The standard deviation of each element is as follows ($SD_i$: *i* indicates individual element). $SD_O < 0.44$, $SD_{Ti} < 0.44$, $SD_{Ca} < 0.12$, $SD_{Sr} < 0.11$, $SD_{Ag} < 0.08$.

The surface structure of these samples was examined by XRD analysis and Raman scattering as shown in Figure 1. Sodium hydrogen titanate (SHT; $Na_xH_{2-x}Ti_3O_7$) was produced after the initial NaOH treatment. These XRD and Raman profiles were not apparently changed except for a slight shift of about 920 to 900 cm$^{-1}$ in Raman by the subsequent Ca + Sr treatment. Since the Raman peak around 920 cm$^{-1}$ in SHT was attributed to Ti–O bonds coordinated with Na ions [35], the results indicate that

the SHT transformed into Sr-containing calcium hydrogen titanate by replacing Na with Ca and Sr without any apparent change of its structural frame. This material was dehydrated by the subsequent heat treatment to form Sr-containing calcium titanate and rutile accompanied by a small quantity of anatase. Although the XRD and Raman profiles were apparently unchanged by the final Sr + 1 mM Ag treatment regardless of the pH of the solution, it may be inferred that the Sr-containing calcium titanate transformed into Sr- and Ag-containing calcium titanate or Sr- and Ag-containing calcium-deficient calcium titanate by a final solution treatment with a pH ≥ 5 or pH ≤ 4, respectively, according to surface chemical composition, as shown in Table 2. When these samples were subjected to scratch resistance test, the surface layer formed after the first NaOH treatment showed low scratch resistance value as 0.9 ± 0.5 mN. This value was almost unchanged by the second solution treatment (the value was 1.6 ± 0.5 mN). In contrast, it markedly increased to 37.8 ± 7.0 mN after heat and remained even after the Sr + 1 mM Ag (pH = 4) treatment (the value was 39.3 ± 3.7 mN).

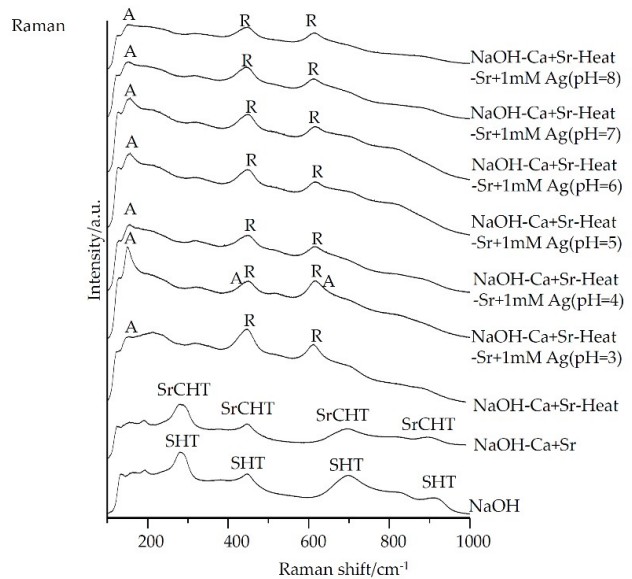

T: α-Ti   A: Anatase   R: Rutile   SHT: $Na_xH_{2-x}Ti_3O_7$   SrCHT: $Sr_xCa_yH_{2-2(x+y)}Ti_3O_7$
SrCT: $Sr_xCa_{1-x}Ti_4O_9$, $Sr_xCa_{1-x}Ti_2O_4$, $Sr_xCa_{1-x}Ti_4O_5$,
▼: Sr- and Ag-containing calcium titanate ($Sr_xAg_yCa_{1-(x+0.5y)}Ti_4O_9$, $Sr_xAg_yCa_{1-(x+0.5y)}Ti_2O_4$, $Sr_xAg_yCa_{1-(x+0.5y)}Ti_4O_5$)

(**a**)

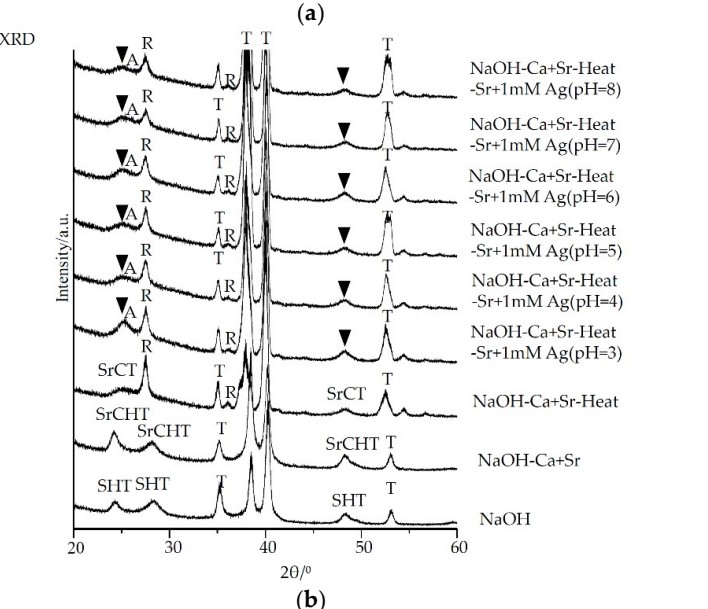

(**b**)

**Figure 1.** Raman (**a**) and XRD (**b**) spectra of Ti surfaces subjected to Sr + 1 mM Ag treatment with various pH following NaOH, Ca + Sr and heat treatment.

Depth profile of XPS analysis on the metal sample after the NaOH-Ca + Sr-heat-Sr + 1 mM Ag (pH = 4) is shown in Figure 2. Comparable amounts of Sr and Ca and a small amount of Ag were detected near the top surface and decreased gradually in depth until approximately 1 μm. The results are consistent with the surface chemical composition in Table 2, and the thickness of the surface layer on cross sectional SEM observation, where an approximately 1 μm thick surface layer was evident (data not shown). Figure 3 shows narrow XPS spectra of the treated metal. The peaks at 367.7 and 373.8 eV attributed to $Ag_2O$ [36] were observed, verifying that Ag was incorporated into the surface as a form of $Ag^+$ ion.

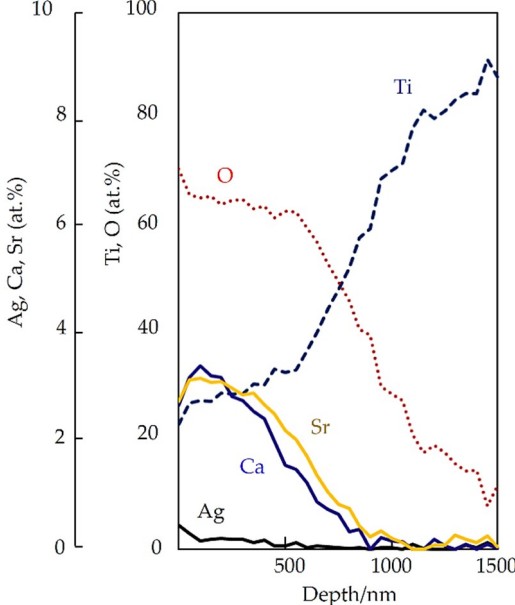

**Figure 2.** XPS depth profile of the surface of Ti subjected to NaOH-Ca + Sr-heat-Sr + 1 mM Ag (pH = 4).

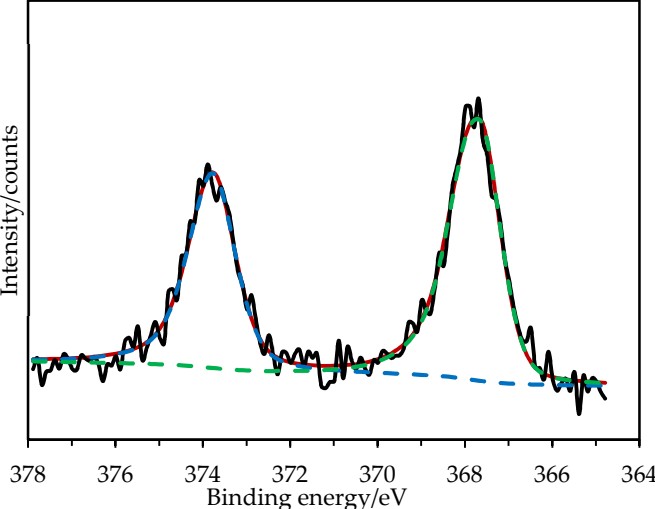

**Figure 3.** Narrow XPS Ag 3*d* profile of Ti surface subjected to NaOH-Ca + Sr-heat-Sr + 1 mM Ag (pH = 4). Red solid line: composite line of blue and green dot lines, Blue dot line: deconvolution line of Ag 3$d_{3/2}$, Green dot line: deconvolution line of Ag 3$d_{5/2}$.

Figure 4 shows the SEM images of the Ti surface before and after soaking in SBF for 3 days that was subjected to Sr + 1 mM Ag treatment with various pH levels following NaOH, $CaCl_2$, and heat treatment. It can be seen that a similar network morphology on a nanometer scale was produced by the aqueous solution and heat treatment regardless of the pH of the 1 M $Sr(NO_3)_2$ + 1 mM $AgNO_3$

solution used in the final solution treatment. When the treated metals were immersed in SBF for 3 days, apatite formation was observed only on the surfaces that had been treated with 1 M Sr(NO$_3$)$_2$ and 1 mM AgNO$_3$ with a pH equal to or less than 4. In terms of apatite formation as well as the Ag and Sr content, the pH of the aqueous solution was fixed at 4 in the following experiment.

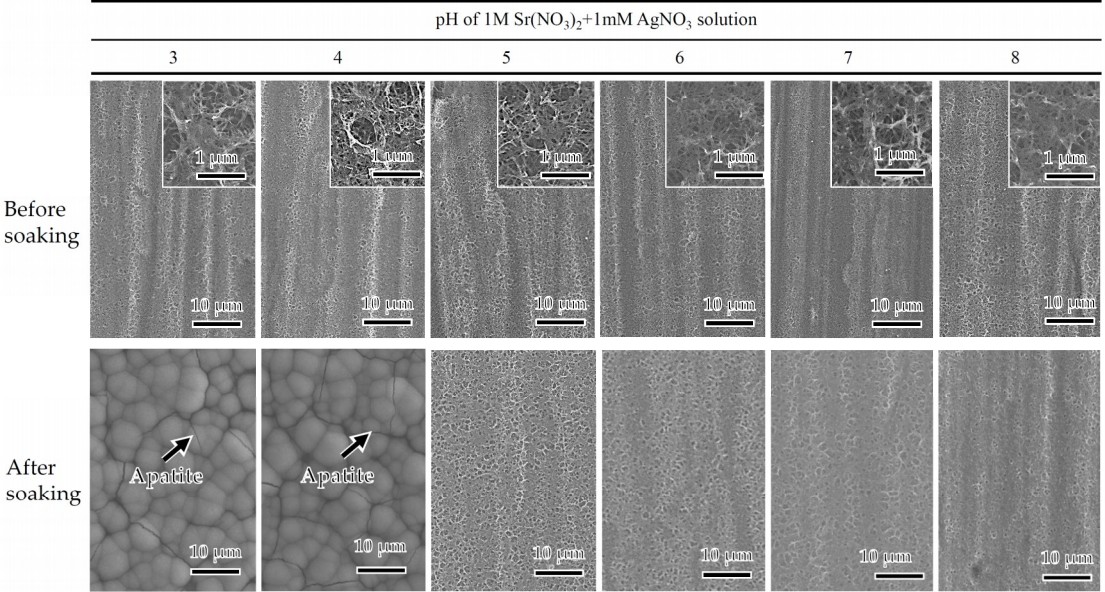

**Figure 4.** SEM images of the surfaces of Ti before and after soaking in SBF for 3 days that has been subjected to Sr + 1 mM Ag treatment with various pH following NaOH, Ca + Sr and heat treatment. Small windows show high magnification images on Ti before soaking in SBF. The digits in the table above the images stand for pH of 1 M Sr(NO$_3$)$_2$ + 1 mM AgNO$_3$ solution in the final solution treatment.

## 3.2. Effect of the Ag Concentration in the Solution Treatment on Apatite Formation

The Ti samples subjected to the NaOH-Ca + Sr-heat treatment were soaked in a 1 M Sr(NO$_3$)$_2$ solution of pH = 4 with different concentrations of AgNO$_3$ from 1 to 100 mM added, and their surface chemical composition was analyzed by EDX, as shown in Table 3. The amount of Ag increased, with increasing concentration of Ag in the final solution treatment up to 1.1% along with slightly decreased Sr content.

**Table 3.** The results of EDX analysis on the surface of Ti subjected to Sr + *X* mM Ag (pH = 4) treatment (*X* = 1–100) following NaOH, Ca + Sr and heat treatment.

| Treatment | Element/at.% | | | | |
|---|---|---|---|---|---|
| | O | Ti | Ca | Sr | Ag |
| NaOH-Ca + Sr-heat-Sr + 1 mM Ag (pH = 4) | 65.8 | 30.5 | 1.9 | 1.6 | 0.2 |
| NaOH-Ca + Sr-heat-Sr + 2 mM Ag (pH = 4) | 66.1 | 30.3 | 1.9 | 1.5 | 0.2 |
| NaOH-Ca + Sr-heat-Sr + 5 mM Ag (pH = 4) | 66.1 | 30.3 | 1.9 | 1.5 | 0.3 |
| NaOH-Ca + Sr-heat-Sr + 10 mM Ag (pH = 4) | 66.3 | 30.2 | 1.8 | 1.4 | 0.3 |
| NaOH-Ca + Sr-heat-Sr + 20 mM Ag (pH = 4) | 66.1 | 30.2 | 1.9 | 1.4 | 0.4 |
| NaOH-Ca + Sr-heat-Sr + 50 mM Ag (pH = 4) | 66.1 | 29.6 | 2.0 | 1.5 | 0.9 |
| NaOH-Ca + Sr-heat-Sr + 100 mM Ag (pH = 4) | 65.7 | 29.9 | 2.0 | 1.3 | 1.1 |

The standard deviation of each element is as follows (SD$_i$: *i* indicates individual element). SD$_O$ < 0.6, SD$_{Ti}$ < 0.4, SD$_{Ca}$ < 0.1, SD$_{Sr}$ < 0.1, SD$_{Ag}$ < 0.1.

SEM revealed that nano sized particles started to be precipitated on the surface of Ti when the Ag concentration in the final solution treatment was 20 mM, and their number increased with an increasing Ag concentration, as shown in Figure 5. These particles were determined to be metallic Ag

particles by EDX line analysis (data not shown). It can be seen from the XRD and Raman spectra of the treated samples (Figure 6) that a peak at around 44° attributed to metallic Ag [37] was detected on the Ti surface only when the Ag concentration in the final solution treatment was equal or greater than 20 mM. There were no other changes that depended on the Ag concentration of the final solution treatment in the XRD and Raman profiles.

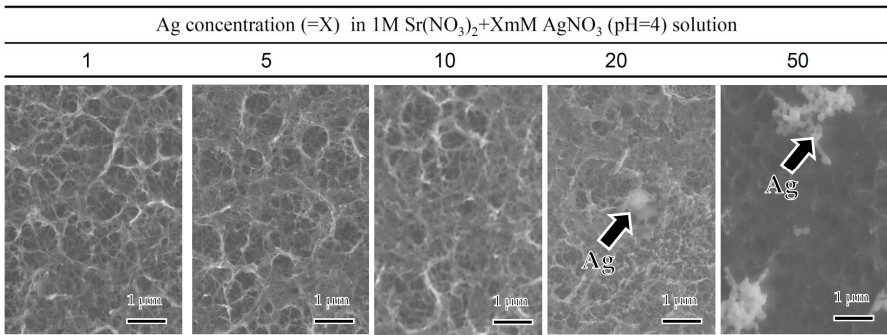

**Figure 5.** SEM images of the surfaces of Ti subjected to Sr + *X* mM Ag (pH = 4) treatment (*X* = 1–50) following NaOH, Ca + Sr and heat treatment.

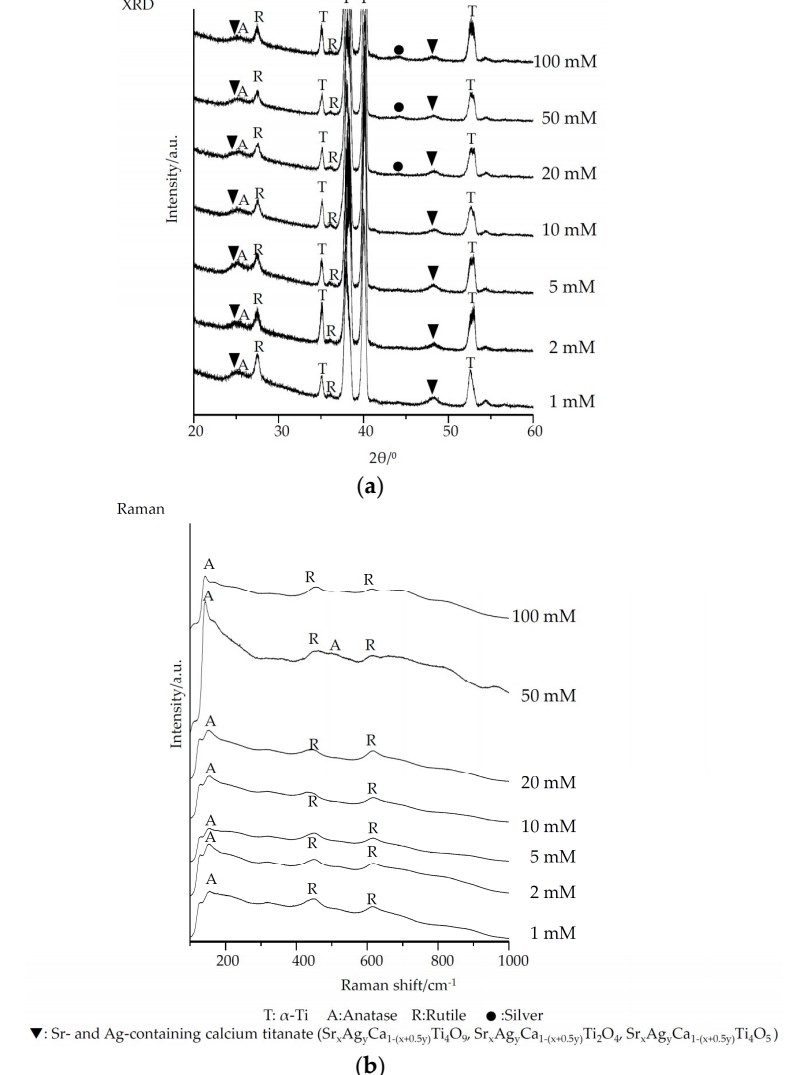

**Figure 6.** XRD (**a**) and Raman (**b**) spectra of Ti surfaces subjected to Sr + *X* mM Ag (pH = 4) treatment with various Ag concentrations (*X* = 1–100) following NaOH, Ca + Sr and heat treatment.

When these samples were soaked in SBF, they formed spherical particles on their surfaces within 3 days that were identified as low crystalline apatite by XRD (data not shown), regardless of the Ag content and even in the presence of Ag particles as shown in Figure 7.

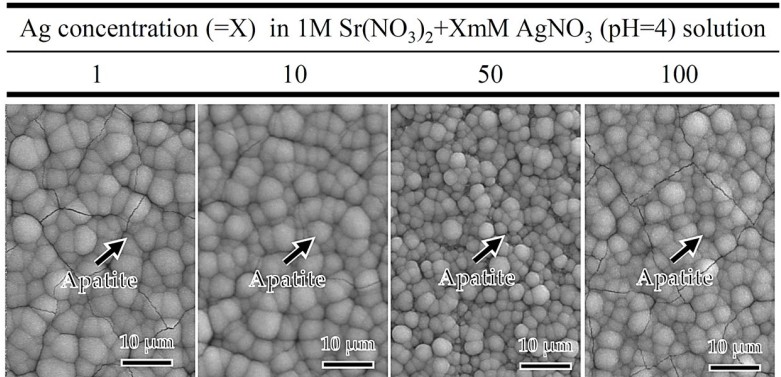

**Figure 7.** SEM images of the surface of Ti before and after soaked in SBF for 3 days that has been subjected to Sr + *X* mM Ag (pH = 4) treatment with various Ag concentrations (*X* = 1–50) following NaOH, $CaCl_2$, and heat treatment.

### 3.3. Effect of the Ag Concentration in the Solution on Cytotoxicity

The Ti samples with Sr- and Ag-containing calcium-deficient calcium titanate without any metallic Ag particles were prepared by Sr + 1 mM Ag (pH = 4) or Sr + 10 mM Ag (pH = 4) treatment following NaOH-Ca + Sr-heat treatment, and their effect on the viability of MC3T3-E1 cells was examined. The results were compared with those on untreated or NaOH-Ca + Sr-heat-Sr-treated Ti with Sr-containing calcium titanate free of Ag. As shown in Figure 8, the cell viability significantly increased in the treated Ti subjected to NaOH-Ca + Sr-heat-Sr + 1 mM Ag (pH = 4) compared with untreated samples in the culture period of 1 day. There were no significant differences between the treated samples. At 3 days, although all of the treated samples showed higher cell viability than the untreated sample, there was a difference between the treated samples: NaOH-Ca + Sr-heat-Sr was highest, followed by NaOH-Ca + Sr-heat-Sr + 1 mM Ag (pH = 4) and then NaOH-Ca + Sr-heat-Sr + 10 mM Ag (pH = 4). There a significant difference between Ti samples subjected to the NaOH-Ca + Sr-heat-Sr and NaOH-Ca + Sr-heat-Sr + 10 mM Ag (pH = 4) treatments.

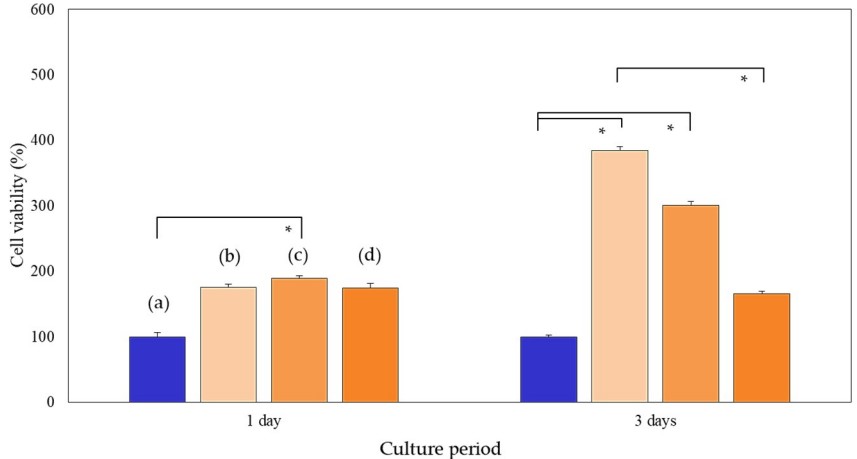

**Figure 8.** Cell viability of MC3T3-E1 on Ti (a) untreated and subjected to (b) NaOH-$CaCl_2$-heat-Sr, (c) NaOH-$CaCl_2$-heat-Sr + 1 mMAg (pH = 4), and (d) NaOH-$CaCl_2$-heat-Sr + 10 mM Ag (pH = 4). Asterisk stands for statistically significant difference ($p < 0.05$).

### 3.4. Antibacterial Activity

The antibacterial activity against *E. coli* of the Ti subjected to NaOH-Ca + Sr-heat-Sr + 1 mM Ag (pH = 4) was examined by the film contact method. As a result, the treated Ti displayed a 5.9-log reduction compared with the untreated Ti, as shown in Table 4, indicating sufficiently high antibacterial activity.

**Table 4.** Antibacterial activity results on Ti untreated and subjected to NaOH-Ca + Sr-heat-Sr + 1 mM Ag (pH = 4).

| Treatment | Average of *E. Coli* count/CFU | | Antibacterial Activity Value |
|---|---|---|---|
| | After Inoculation | After Incubation | |
| Untreated | $2.8 \times 10^6$ | $1.5 \times 10^7$ | – |
| NaOH-Ca + Sr-heat-Sr + 1 mM Ag (pH = 4) | $4.7 \times 10^6$ | <20 | 5.9 |

### 3.5. Ion Release Test

The same treated samples were soaked in FBS for up to 14 days and the Sr and Ag ions released from the samples were measured by ICP. It can be seen in Figure 9 that the treated metal released 0.78 ppm of Ag and 0.87 ppm of Sr within 1 h, and then slowly released another 0.91 ppm of Ag and 0.42 ppm of Sr over 14 days.

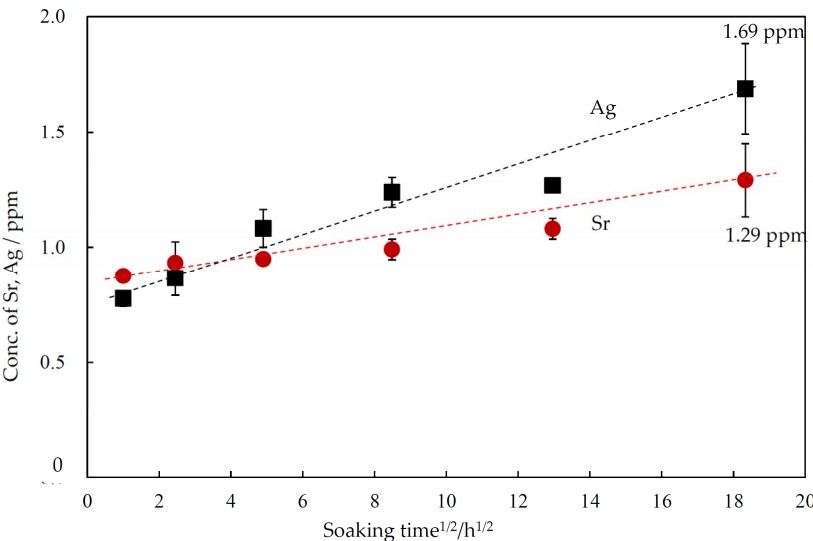

**Figure 9.** Concentrations of Sr and Ag ions released from Ti subjected NaOH-CaCl$_2$-heat-Sr + 1 mM Ag (pH = 4), as a function of square root of soaking time in FBS, which was measured by ICP. Black squares and red circles indicate Ag and Sr, respectively.

## 4. Discussion

The SHT (Na$_x$H$_{2-x}$Ti$_3$O$_7$) formed on Ti and its alloys by NaOH treatment has potent ion exchange capacity because of its layered structure [38]. It has been reported that the Na$^+$ in SHT can be exchanged by various types of and even various valences of metal ions such as Ag$^+$, magnesium (Mg$^{2+}$), Ca$^{2+}$, Sr$^{2+}$, gallium (Ga$^{3+}$), and more than two types of them simultaneously [27–30,39–42]. The present study proved that even three types of functional ions such as Ca, Sr and Ag can be controllably incorporated into the surface of Ti by a combination of simple aqueous solution and heat treatment.

The incorporation of Ag$^+$ ions into SHT was first attempted by Inoue et al. [43]. They formed an Ag-containing sodium titanate layer by soaking Ti in 0.05 M silver acetate solution following NaOH solution or NaOH hydrothermal treatment. The treated metal contained Ag$^+$ ions, but also a certain amount of precipitated metallic silver particles [43]. Such colloidal metallic particles are undesirable because they could be released and transferred to other organ, resulting in side effects.

Kizuki et al. [28] and Prabu et al. [44] demonstrated that Ag can be induced into SHT formed on Ti and Ti–6Al–4V alloy as ion form by soaking the metals in 0.01–100 mM $Ag(NO)_3$ solution after the NaOH solution. However, metallic Ag particles were formed again when the treated metals were subsequently heat-treated at 600 °C [28]. Eventually, Ag-containing calcium titanate free of Ag particles forms on Ti and Ti–15Zr–4Nb–4Ta alloy as the result of immersing in 1 mM $AgNO_3$ solution following NaOH-CaCl$_2$-heat treatment [28]. The treated Ti and Ti–15Zr–4Nb–4Ta contained 0.55% and 0.27% of Ag on their surfaces and released 2.66 and 1.45 ppm of $Ag^+$ ions into FBS, respectively. These metals showed strong antibacterial activity against *S. aureus* (more than 99% reduction). However, the cytotoxicity of these treated metals was not reported.

In the present study, Ti along with Sr- and Ag-containing calcium titanate with various amounts of Ag were produced. It is shown in Table 3 and Figures 5 and 6 that the Ag content increased from 0.2% to 1.1% in the final solution treatment, while metallic Ag precipitated when the Ag content on the surface of Ti became 0.4%. Thus, Ti specimens with the surface containing 0.2% and 0.3% Ag in the ion form were prepared and their cytotoxicity compared with the untreated and the Ag free Ti with the Sr-containing calcium titanate. As a result, the Ti with the Sr-containing calcium titanate was shown to markedly increase the cell viability after 3 days compared with the untreated Ti. This is consistent with our previous report in which the cell viability in the Ti subjected to the same treatment was significantly increased compared with untreated Ti on 5 days [45]. A similar increase in cell viability was observed in the case of Ti with 0.2% Ag at 1 and 3 days. In contrast, the cell viability of Ti with 0.3% Ag was significantly lower than that of Ti with Sr-containing calcium titanate, although it was comparable to that of untreated Ti. This is probably because the $Ag^+$ ions that were released from the surface of the treated Ti accumulated in the vicinity of the surface and suppressed cell proliferation. The Ti with the 0.2% Ag resulting from NaOH-Ca + Sr-heat-Sr + 1 mM Ag (pH = 4) slowly released 1.69 ppm $Ag^+$ ions into FBS over 14 days and exhibited potent antibacterial activity against *E. coli*, as shown in Figure 9 and Table 4. It has been reported that Ti and its alloys with 0.27%–0.67% Ag content induced by chemical and heat treatment exhibited strong antibacterial activity without any cytotoxicity [46]. The results in this study are consistent with these reports. On the other hand, they also imply that a lower Ag content, such as 0.3%, may suppress cell proliferation without any cytotoxicity.

Bone-bonding is a crucial function of an implant and may be predicted by examining the apatite formation on the material that occurs in SBF [33]. Fujibayashi et al. reported that bioactive Na$_2$O-CaO-SiO$_2$ glass powders with different compositions and induction periods of apatite formation in SBF induced different amount of new bone formation: amount of new bone formation increased with decreasing induction periods of apatite formation in SBF [47]. They recommended the materials able to form apatite within 3 days in SBF for practical use. It should be noted that the bone-bonding strength might be affected by various factors including strength and thickness of the substrate and coating layers [48]. It was reported that Ti–15Zr–4Nb–4Ta alloy with approximately 0.5 μm calcium-deficient calcium titanate layer implanted into rabbit tibia showed lower critical detaching load in detaching test than Ti with approximately 1 μm calcium-deficient calcium titanate layer at 4 weeks of implantation period, although both of them exhibited direct bone-bonding in histological observation [19,20]. The critical detaching load increased with increasing implantation periods up to 26 weeks in both cases, where fracture occurred not at interfaces between the treated metals and bone but inside the bone [20]. In this study, abundant apatite formation was observed on the treated Ti within 3 days in SBF regardless of the Ag content and even in the presence of metallic Ag particles. Sufficiently high bone-bonding is expected on these metals. On the other hand, apatite formation strongly depended on the pH of the solution in the final solution treatment: Ti formed apatite only when it had been soaked in 1 M Sr(NO$_3$)$_2$ and 1 mM AgNO$_3$ with a pH equal to or less than 4, as shown in Figure 7. This might be due to the formation of Sr- and Ag-containing calcium-deficient calcium titanate on Ti. It is reported that the calcium-deficient calcium titanate that forms on Ti exhibits an increased capacity for apatite formation compared with calcium titanate because of its greater release of $Ca^{2+}$ ions [28,41].

It is expected that new bone growth surrounding Ti will be accelerated if appropriate concentrations of $Sr^{2+}$ ions are released from Ti in the living body. The new bone tightly bonds to the metal via the apatite that have been formed on the metal surface. In the present study, the Ti subjected to NaOH-Ca + Sr-heat-Sr + 1 mM Ag (pH = 4) slowly released 1.29 ppm of $Sr^{2+}$ ions into FBS in addition to $Ag^+$ ions over 14 days. This value falls in the effective range of 0.21 and 21.07 ppm that was shown to enhance the expression of a key osteoblast transcription factor gene (Cbfa1) and alkaline phosphatase (ALP) activity in human bone marrow mesenchymal stem cells [49]. Park et al. hydrothermally produced $SrTiO_3$ coating on Ti that released 0.75 ppm of $Sr^{2+}$ ions into physiological saline solution. When primary mouse bone marrow stromal cells were cultured on the metal surface, increased cell activity including cell attachment, spreading, gene expression, and ALP activity was shown [30]. Yamaguchi et al. [29] and Okuzu et al. [45] reported that Ti having Sr-containing calcium titanate on its surface increased proliferation and osteogenic differentiation of MC3T3-E1 cells by releasing 0.92 ppm of $Sr^{2+}$ ions. In their reports, various types of gene expression, including integrin β1, β catenin, cyclin D1 and ALP were up-regulated and resulted in extracellular mineralization. They also showed that biomechanical strength as well as bone-implant contact became greater than the Ti with calcium-deficient calcium titanate, when the metals were implanted into rabbit tibia at short periods of 4–8 weeks.

Based on these results, the Ti with Sr- and Ag-containing calcium-deficient calcium titanate is expected to form apatite on its surface and bond to living bone through the apatite, while promoting new bone growth by releasing $Sr^{2+}$ ions. Furthermore, it should prevent postoperative infection because of its antibacterial activity.

## 5. Conclusions

Tri-functional bioactive Ti with the Sr- and Ag-containing calcium-deficient calcium titanate was produced by a combination of aqueous solution and heat treatment. An effective amount of Ca, Sr and Ag was introduced into the surface of Ti by controlling the ion concentration and pH of the solution so that the treated Ti precipitated apatite in SBF within 3 days and exhibited strong antibacterial activity, with increased cell viability. Furthermore, it released $Sr^{2+}$ ions into FBS at a level up to 1.29 ppm. This type of multifunctional Ti is promising for the next generation of orthopedic and dental implants in next generation.

**Author Contributions:** Conceptualization, S.Y., M.I., S.A.S. and H.T.; Methodology, S.Y., P.T.M.L. and M.I.; Software, S.Y. and S.A.S.; Validation, S.Y., P.T.M.L., M.I. and S.A.S.; Formal Analysis, S.Y., M.I. and S.A.S.; Investigation, S.Y., P.T.M.L. and M.I.; Resources, S.Y. and M.I.; Data Curation, S.Y. and S.A.S.; Writing—Original Draft Preparation, S.Y.; Writing—Review and Editing, S.Y.; Visualization, S.Y.; Supervision, M.I. and H.T.; Project Administration, S.Y.; Funding Acquisition, S.Y.

**Funding:** This research was partially supported by Chubu University Grant (B) 19M02B.

**Conflicts of Interest:** The authors declare no conflict of interest.

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
