# Peer review of "Tri-Functional Calcium-Deficient Calcium Titanate Coating on Titanium Metal by Chemical and Heat Treatment"

_coatings, doi:10.3390/coatings9090561_

Round 1
Reviewer 1 Report
The paper describes a new method for the creation of titanium (Ti) implants for bone healing.
The paper is excellent and I recommend acceptance subject to some minor corrections listed below.
1. Throughout the text italicize Escherichia coli (E. coli)
2. page 1, line 31-32 - "at several to several dozen micro meters by plasma spraying" Please be more specific here
3. page 1, line 32 - Please explain "increase cell activity". Does this mean proliferation of the cells, cellular metabolic activity, etc.?
4. page 1, line 33 - Rewrite "but still do bond to it adequately" as "but still bond to it adequately"
5. I would recommend two separate figures for the Raman and XRD results as they are difficult to see combined.
6. In Figure 3 No. 3,4,5,6, etc at the top of the pictures are the pH levels?
7. In the caption to Figure 8 add a line saying black squares are Ag and red circles are Sr
Author Response
Dear Reviewer,
We appreciate your valuable comments, and would like to reply them as follows.
1.The paper describes a new method for the creation of titanium (Ti) implants for bone healing. The paper is excellent and I recommend acceptance subject to some minor corrections listed below.
We appreciate you for your valuable comments.
2. Throughout the text italicize Escherichia coli (E. coli)
Thank you very much for your comment. We italicized the words throughout the text.
3. page 1, line 31-32 - "at several to several dozen micro meters by plasma spraying" Please be more specific here
Thank you for your valuable comment. Increased osseointegration was reported at micro meter scale such as 0.5 – 2.0, or 3.6 -5.6 μm in Ra, and 43 – 50 μm in Rz by plasma spraying, grid blasting and/or acid etching [refs 1,2]. This information was added to our manuscript.
4. page 1, line 32 - Please explain "increase cell activity". Does this mean proliferation of the cells, cellular metabolic activity, etc.?
Thank you for your important comment. It was reported that the nanometer scale roughness produced by anodic oxidation increased cell adhesion, proliferation and alkaline phosphatase activity [refs 3,4]. This information was added to our manuscript.
5. page 1, line 33 - Rewrite "but still do bond to it adequately" as "but still bond to it adequately"
Thank you for your valuable comment. The sentence contains a mistake. It was rewrote as following. “but still do not bond to it adequately” in our revised manuscript.
6. I would recommend two separate figures for the Raman and XRD results as they are difficult to see combined.
Thank you for your valuable comment. The figures of Raman and XRD results were rearranged to be easily viewable.
7. In Figure 3 No. 3,4,5,6, etc at the top of the pictures are the pH levels?
Yes it is. The explanation was added in the figure caption.
8. In the caption to Figure 8 add a line saying black squares are Ag and red circles are Sr
Thank you for your valuable comment. The explanation was added in the figure caption.
Reviewer 2 Report
1. The authors report antibacterial effect from Ti-based Sr and Ag Ca-deficient coating. However, more in-depth analysis with the literature data should be provided, for example with the other most frequently used silver-containing compounds (Materials Science and Engineering: C, Volume 97, 2019, Pages 420-430; Colloids and Surfaces B: Biointerfaces, Volume 156, 2017, Pages 104-113)
2. The authors should compare different results relative to the thickness of the coating prepared. Because, the thickness will define cell adhesion and antibacterial effect observed. How correspond the degradation rate of the deposited coatings with the bone formation rate. Did heat treatment after coating deposition deteriorate films adhesion strength?
Author Response
1. The authors report antibacterial effect from Ti-based Sr and Ag Ca-deficient coating. However, more in-depth analysis with the literature data should be provided, for example with the other most frequently used silver-containing compounds (Materials Science and Engineering: C, Volume 97, 2019, Pages 420-430; Colloids and Surfaces B: Biointerfaces, Volume 156, 2017, Pages 104-113)
Thank you very much for your important comments. XPS high-resolution analysis was performed on Ti subjected to NaOH-Ca+Sr-heat-Sr+1mM Ag (pH=4). As a result, it was found that the peaks at 367.7 and 373.8 eV attributed to Ag2O were observed, indicating that Ag was incorporated into the surface as a form of Ag+ ion. The result was added in our manuscript.
2. The authors should compare different results relative to the thickness of the coating prepared. Because, the thickness will define cell adhesion and antibacterial effect observed. How correspond the degradation rate of the deposited coatings with the bone formation rate.
Thank you very much for your important comments. The thickness and degradation rate of the coatings may effect cell adhesion and bone formation if the coatings are degradable and have poor apatite formation. We would like to insist that the calcium deficient calcium titanate doped with Sr and Ag in the present manuscript has insoluble TiO2 frame structure although it can release some amount of Ca. The treated metal formed apatite fully on its surface within 3 days in SBF. This apatite-forming capability is high enough to exhibit bone bonding according to the previous report that mentions relationship between apatite formation and bone bonding [R1]. On the other hand, bonding strength between bone and implant might be influenced by various factors such as strength of the substrate and thickness and strength of the coating layers [R2]. It was reported that Ti-15Zr-4Nb-4Ta alloy with approximately 0.5 μm calcium-deficient calcium titanate layer implanted into rabbit tibia showed lower critical detaching load in detaching test than Ti with approximately 1 μm calcium-deficient calcium titanate layer at 4 weeks, although both of them exhibited direct bone bonding in histological observation [19,20]. The critical detaching load increased with increasing implantation periods up to 26 weeks in both cases, where fracture occurred not at interface between the treated metals and bone but inside bone [20].
These explanation and the following references were added to the revised manuscript.
[R1] Fujibayashi, S.; Neo, M.; Kim, H.-M.; Kokubo, T,; Nakamura, T. A comparative study between in vivo bone ingrowth and in vitro apatite formation on Na2O–CaO–SiO2 glasses. Biomaterials 2003, 24, 1349–1356.
[R2] Takemoto, M.; Nakamura, T. In Bioceramics and their clinical applications; Kokubo, T., Ed.; Woodhead Publishing: Cambridge, UK, 2008; Chapter 8, pp 183-198.
3. Did heat treatment after coating deposition deteriorate films adhesion strength?
Thank you very much for your important question. Scratch resistance test was performed on Ti subjected to NaOH-Ca+Sr-heat-Sr+1mM Ag (pH=4) according to the reviewer’s comment. As a result, the scratch resistance of the surface layer formed by initial NaOH treatment showed low value as 0.9 ± 0.5 mN. This value was almost unchanged by the second solution treatment (the value was 1.6 ± 0.5 mN). In contrast, it markedly increased to 37.8 ± 7.0 mN after heat treatment and remained even after the Sr+1mM Ag (pH=4) treatment (the value was 39.3 ± 3.7 mN) .
These results were added to the revised manuscript.
Round 2
Reviewer 2 Report
The author did not sufficiently compared the results obtained with those reported in the literature, i.e. comment 1. After this point is properly addressed the manuscript can be considered for publication.
Author Response
1. The author did not sufficiently compared the results obtained with those reported in the literature, i.e. comment 1. After this point is properly addressed the manuscript can be considered for publication.
We are sorry to misunderstand your comment. The following references are added to our revised manuscript [31,32] to compare the results.
[31] Chernozema, R.V.; Surmeneva, M.A.; Krauseb, B.; Baumbach, T.; Ignatov, V.P; Prymak, O.; Loza, K.; Epple, M.; Ennen-Roth, F.; Wittmar, A.; Ulbricht, M.; Chudinova, E.A.; Rijavecf, T.; Lapanjef, A.; Surmenev, R.A. Functionalization of titania nanotubes with electrophoretically deposited silver and calcium phosphate nanoparticles: Structure, composition and antibacterial assay. Mater. Sci. Eng. C 2019, 97, 420-430.
[32] Surmeneva, M.A.; Sharonova, A.A.; Chernousova,S.; Prymak, O.; Loza, K.; Tkachev, M.S.; Shulepov, I.A.; Epple, M.; Surmenev, M.A. Incorporation of silver nanoparticles into magnetron-sputtered calcium phosphate layers on titanium as an antibacterial coating. Colloids Surf. B: Biointerfaces 2017, 156, 104-113,